# Investigation of Phase Transitions in Ferromagnetic Nanofilms on a Non-Magnetic Substrate by Computer Simulation

**DOI:** 10.3390/ma15072390

**Published:** 2022-03-24

**Authors:** Sergey V. Belim

**Affiliations:** Physics Department, Omsk State Technical University, Omsk 644050, Russia; sbelim@mail.ru

**Keywords:** thin film, nonmagnetic substrate, phase transition, computer simulation, Frenkel-Kontorova potential

## Abstract

Magnetic properties of ferromagnetic nanofilms on non-magnetic substrate are examined by computer simulation. The substrate influence is modeled using the two-dimensional Frenkel-Kontorova potential. The film has a cubic crystal lattice. Cases of different ratio for substrate period and ferromagnetic film period are considered. The difference in film and substrate periods results in film deformations. These deformations result in a change in the magnetic properties of the film. The Ising model and the Metropolis algorithm are used for the study of magnetic properties. The dependence of Curie temperature on film thickness and substrate potential parameters is calculated. Cases of different values for the coverage factor are considered. The deformation of the film layers is reduced away from the substrate when it is compressed or stretched. The Curie temperature increases when the substrate is compressed and decreases when the substrate is stretched. This pattern is performed for films with different thicknesses. If the coating coefficient for the film is different from one, periodic structures with an increased or reduced concentration of atoms are formed in the film first layer. These structures are absent in higher layers.

## 1. Introduction

Ferromagnetic film composite systems on a non-magnetic substrate are widely used in spintronics devices. A substrate of non-magnetic material has no effect on the giant magnetic resistance. The ability to control magnetization in such a system is important. Antiferromagnetic or ferromagnetic substrates change the state of the magnetic film, creating an exchange bias. This effect is nonlinear. Non-magnetic substrates can change the mechanical properties of the film. Magnetostrictive effects change the magnetization of the film during mechanical deformations. The substrate deformations can cause a ferromagnetic phase transition at temperatures below the Curie point for the free film.

The influence of the substrate on various physical properties (optical, electrical and magnetic) in epitaxial films has been experimentally studied [1,2,3,4,5]. The study of the substrate deformations effect on the magnetic properties in thin ferromagnetic films is of great interest. Film deformations may be caused by thermal expansion of the substrate [6,7]. The substrate may be made of ferroelectric material. In this case, compression or tension deformations can be caused by an electric field [8,9,10]. We’re looking at non-magnetic substrates. The effects of ferromagnetic and antiferromagnetic substrates require consideration of additional exchange and dipole–dipole interactions.

Different types of interactions may occur on the film and substrate interface. Chemical bonds at the interface between the film and the substrate play a large role in the formation of the film by various techniques. These bonds can determine the type of crystal lattice for the film. One type of crystal lattice is studied in this work; therefore, the main type of interaction by Van der Waals forces is considered. Different types of interactions should be considered when studying structural phase transitions induced by the substrate in nanofilms.

The effect of a non-magnetic substrate on epitaxial films is due to the interaction of its atoms with the film atoms. A change in the mutual arrangement of atoms occurs when the substrate is deformed. The film atoms are displaced due to interaction with the substrate atoms. Deformation of the film occurs due to displacement of its atoms. Magnetostrictive effects lead to a change in the magnetic properties of the film. The dependence of the exchange integral for the spin–spin interaction on distance has a main influence on the change in magnetic properties. Studies of the substrate deformations effect on the magnetic properties of single-layer 2D films on a non-magnetic substrate [11,12,13] have shown that uniform substrate deformations can lead to non-uniform film deformations at the free ends of the film. Islands with increased concentration of atoms are formed in the film when the substrate is compressed. These islands are separated by a periodic structure with a reduced concentration of atoms. Islands with reduced concentration of atoms are formed when the substrate is stretched. The space between the islands is a periodic superstructure with an increased concentration of atoms. If it is limited to the case with the exponential law for changing the exchange interaction with distance, then in both cases the Curie temperature of the ferromagnetic film decreases. If the exchange interaction decreases with the distance according to a slower law, then with a small compression of the substrate, the Curie temperature can increase. These patterns are qualitatively consistent with the experiment’s results [14,15,16].

We can expect a more complex dependence for magnetic properties when deforming a substrate for films with a thickness of more than one layer. Curie temperature depends on film thickness at any substrate [17,18]. The non-magnetic substrate interacts only with the film atoms at the boundary of their contact. The substrate does not directly affect the second and subsequent film layers due to the rapid decrease in intermolecular interaction with distance. The layer of film atoms directly contacting the substrate is influenced by the substrate and the subsequent atoms layer. Film deformations are complex. The deformation of the film layers is reduced when removed from the substrate. We can expect that an increase in film thickness will reduce the effect of the substrate on the film magnetic properties.

Computer simulation of a thin ferromagnetic film on a substrate with a periodic potential is performed in this article. The simulation was carried out in two stages. In the first step, the size change for the film layers is examined by compressing and stretching the substrate. In the second step, the change in the magnetic properties of the film is studied under the influence of these deformations.

## 2. Model and Methods

The substrate influences the arrangement of the ferromagnetic film atoms due to the interaction between their atoms. We are studying the case for a substrate with a cubic crystal lattice. Atoms in such a substrate create a periodic potential for film atoms. This potential can be written as a two-dimensional Frenkel–Kontorova potential [19].
(1)$$U_{sub}=\frac{A}{2}\sum_{n}\left(1-\cos\left(\frac{2\pi }{b}x_{n}\right)\times \cos\left(\frac{2\pi }{b}y_{n}\right)\right)$$
*b* is the substrate crystal lattice period, *A* is the substrate potential amplitude and (*x_n_*, *y_n_*) are the film atom coordinates at the border with the substrate.

Interatomic interaction rapidly decreases with distance. Van der Waals forces obey the power law *r*^−6^. The substrate has a direct effect only on the film layer that contacts it. This first atomic film layer can deform and cause deformations in the following layers. Harmonic approximation is used to describe the interaction energy between atoms in a film.
(2)$$U_{\mathrm{int}}=\frac{g}{2}\sum_{n}\left({\left(x_{n+1}-x_{n}-a\right)}^{2}+{\left(y_{n+1}-y_{n}-a\right)}^{2}+{\left(z_{n+1}-z_{n}-a\right)}^{2}\right)$$
*g* is an elastic constant and *a* is the period of the crystal lattice for the undeformed film.

We are looking at a cubic lattice film. The substrate’s potential depends only on the *x* and *y* coordinates. We do not consider film deformations perpendicular to the OXY plane.

Each ferromagnetic film atom is described by the position in space (*x_n_*, *y_n_*, *z_n_*) and spin *S_n_*. We use the Ising model to describe the film’s magnetic properties. The spin can take one of two values (1/2 or −1/2) in this model. Magnetic phenomena in the film are described by Hamiltonian of the Ising model.
(3)$$H=-\sum_{i,j}J\left(|{\vec{r}}_{i}-{\vec{r}}_{j}|\right)S_{i}S_{j}$$

The exchange integral *J* depends on the distance between the spins. The law of this dependence can be different for different substances. The exchange integral rapidly decreases with distance in most substances. This case can be described by exponential law.
(4)$$J\left(|{\vec{r}}_{i}-{\vec{r}}_{j}|\right)=J_{0}\exp\left(-\left(|{\vec{r}}_{i}-{\vec{r}}_{j}|-a\right)/r_{0}\right)$$

The parameter *r*_0_ determines the speed of energy decrease for exchange interaction between spins with distance. Summing only on the nearest neighbors remains in Hamiltonian for the Ising model in this case. The descending of the exchange integral according to a slower law leads to long-range effects. Long-range effects correct critical behavior near phase transition line [20].

It is necessary to minimize the total energy for the system to find an equilibrium state.
(5)$$U_{sub}+U_{\mathrm{int}}+H\to \min$$

Spin dynamics are characterized by a much shorter relaxation time than the mechanical movement of atoms. Therefore, the task can be divided into two parts. The equilibrium mechanical state for film atoms may be determined in a first step. After that, the spin dynamics can be studied at an unchanged position for atoms. We first minimize the mechanical part of the potential energy.
(6)$$U=U_{sub}+U_{\mathrm{int}}\to \min$$

The atom’s coordinates are used to calculate exchange integrals for the interaction between neighboring spins.

Changing the substrate potential period simulates substrate compression. Film edges are rigidly fixed at substrate boundaries in the considered model. Linear dimensions of the first film layer coincide with linear dimensions of the substrate. Higher layers can resize because they do not interact directly with the substrate.

The sequential approximation method is used to search for an equilibrium state. The film atoms are located at the nodes of the cubic crystal lattice with period *a* in zero approximation. All film atoms are bypassed at each iteration. Each atom is shifted by a random vector (*dx*, *dy*) in the OXY plane. The displacement of the atom does not exceed 0.01*a*. If the new position is more energetically advantageous, then it is accepted, otherwise the atom returns to the original position. The total number of iterations is 10,000. As the computer experiment showed, subsequent iterations do not change the position of atoms. The distribution of atoms in the film in the basic state depends on the ratio of the substrate crystal lattice periods and the undisturbed film b/a and the coating coefficient *θ*.
(7)θ=L/M

*L* is the number of film atoms along one axe and *M* is the number of substrate minima along one axe.

The Wolf cluster algorithm [21] is used to study phase transitions in thin films. The magnetization in the film *m* is used as order parameter for the system.
(8)$$m=\sum S_{i}/N$$
*N* is the number of atoms in the film.

Periodic boundary conditions are used in modeling magnetic phenomena. Finite dimensional scaling theory is used to calculate phase transition parameters [22]. Systems with different linear dimensions are modeled. The behavior for an infinite system is approximated based on the results for the endpoint systems. The fourth order Binder cumulants [23] are calculated to determine the phase transition temperature.
(9)$$U=1-\frac{\langle m^{4}\rangle }{3{\langle m^{2}\rangle }^{2}}$$

Angle brackets denote averaging by thermodynamic configurations. The dependence of Binder cumulants on temperature is calculated for systems with different linear dimensions. The Binder cumulant value does not depend on the size of the system at the phase transition temperature, so all plots must intersect at one point [22]. The Curie temperature *Tc* is calculated from the crossing point of the plots. The temperature of system *T* is measured in relative units *J*_0_/*k_B_* (*k_B_*—Boltzmann constant) in computer modeling.

Crystal lattices with different geometry can be obtained by varying lattice periods along the OX and OY axes in potential (1). Examination of substrates with any crystal lattices can be carried out by adding periodic terms to the potential (1). The algorithm proposed in this paper can be used to model a system with any substrate potential. If the symmetry of the crystal lattice for the film and the substrate is different, then competition of the ordering types is observed in the system.

## 3. Results

Computer experiment performed for films with size *L* × *L* × *D*. The film thickness varies from *D* = 2 to *D* = 8 in increments Δ*D* = 2. A film having a thickness one-layer *D* = 1 is also considered. The undeformed crystal lattice period in the film is *a* = 1 in a computer experiment. Values from *b* = 0.9 to *b* = 1.10 in increments of Δ*b* = 0.05 are selected for the substrate period. The linear dimensions for the system vary from *L* = 20 to *L* = 100 in increments Δ*L* = 10. The exchange integral parameter is *r* = 0.1. The value of *r* affects the rate at which the exchange interaction with distance descends. Increasing this parameter reduces the change in phase transition temperature during substrate deformation. The general patterns remain unchanged. Substrate potential amplitude *A* = 0.5 was used in the simulation. Increasing the substrate potential amplitude slightly changes the phase transition temperature. Decreasing the substrate potential amplitude brings the Curie temperature closer to the undeformed film temperature.

Systems with a coverage factor *θ* = 1 are considered in the first stage. In this case, the first layer atoms interacting with the substrate are located in the minima of the potential *U_sub_*. These atoms offset changes the linear dimensions of the layer. The relative change in layer size along one direction is indicated through the *ε*.
(10)ε=ΔL/L

The relationship of the relative increase in the layer’s linear length *ε* depending on its number *d* at *b* = 1.10 is shown in Figure 1.

As seen in Figure 1, substrate deformations affect all layers. The substrate influence decreases as the number of the atomic layer increases. The first film layer interacts most intensively with the substrate. The deformations for the first layer almost exactly repeat the deformations for the substrate. Deformations are transmitted to the following layers by elastic forces of interaction between the film atoms. For layers with numbers greater than the first, there is a competition of elastic forces. The first force tends to deform the layer under the influence of the substrate. The second force tends to return the atoms to their original equilibrium state. The first force is created by layers located closer to the substrate. The second force is created by higher layers. The first force decreases as the distance to the substrate increases. Substrate deformations affect each next layer less.

The deformation of the film changes the distances between the spins. This change affects the exchange integral and Curie temperature of the film. A similar pattern is observed for another values *b* ≠ 1. System compression occurs at substrate period *b* < 1. The distance between the spins is reduced. The exchange integral increases. Curie’s temperature is rising. System straining occurs at *b* > 1. The phase transition temperature decreases. A plot for Curie temperature versus substrate period is shown in Figure 2 for films with different thicknesses.

As shown in Figure 2, the phase transition temperature is lowered by stretching the substrate and increased by compressing the substrate. The phase transition temperature increases with increasing film thickness. The general pattern for change in phase transition temperature from substrate deformation is the same for films with different thicknesses.

The dependence of Curie temperature on film thickness at various substrate deformations is shown in Figure 3.

As shown in Figure 3, the Curie temperature increases with increasing film thickness. Phase transition temperature reaches asymptotic value *T*_0_ = 4.51 at *b* = 1.00. This asymptotic value corresponds to the Curie temperature for the three-dimensional Ising model. The asymptotic value increases when the substrate is deformed.

The change in Curie temperature is caused by a change in the exchange integral for the interaction between spins. The exchange integral changes due to a decrease or increase in the distance between atoms possessing spins. The change in Curie temperature can be considered as a manifestation of the magnetostrictive effect during film deformation.

Systems with a coverage factor not equal to one are considered in the second stage. If the coating coefficient is *θ* < 1, then the atom density is less than for a free lattice. This atom arrangement leads to a tighter package. Curie temperature rises. A less dense package of atoms is realized for the coating factor *θ* > 1. Curie temperature drops. The phase transition temperature is further varied by substrate deformation. The main patterns of change in Curie temperature when the substrate period changes remain the same as when the single coating coefficient *θ* = 1 (Figure 4). The law for changing the phase transition temperature with increasing film thickness remains the same as for a single coating coefficient *θ* = 1 (Figure 5).

The deformation’s nature changes with a coating coefficient not equal to one (*θ* ≠ 1). Periodic structures are formed in the first film layer under the influence of the substrate. The shape of these structures depends on the coating coefficient. If the coating coefficient is less than one (*θ* < 1), square regions with an increased concentration of atoms are formed. These regions are separated by spaces with a reduced concentration of atoms (Figure 6a). These structures are weakly manifested in the second layer (Figure 6b). Atoms are located at the nodes of the square lattice in higher layers.

If the coating coefficient is greater than one (*θ* > 1), then square regions with a reduced concentration of atoms are formed in the first layer under the influence of the substrate (Figure 7a). These regions are separated by a space with an increased concentration of atoms. These structures are weakly manifested in the second layer (Figure 7b). These structures are practically untraceable in higher layers.

Periodic structures in the first layer are preserved when the substrate is deformed. The distance between atoms changes. The relative arrangement of atoms remains unchanged.

## 4. Conclusions

Ferromagnetic phase transition in thin multilayer films on a deformable non-magnetic substrate was studied by computer modeling. Substrate deformations change phase transition temperature. The increase in film thickness also increases the phase transition temperature. These facts are consistent both with the experimental data [14,15,16] and with the results of modelling by other methods [17,18]. The film coating coefficient also affects the phase transition temperature. Deformation of the substrate causes general compression or stretching in the film without changing its structure at a single coating coefficient. The layer interacting with the substrate experiences the greatest deformation. The deformations of the remaining layers decrease when removed from the substrate. If the coating coefficient is not one, periodic structures are formed in the layer interacting with the substrate. These structures are islands with increased or decreased concentrations of atoms. These structures are most pronounced in monoatomic films. These structures are weakly expressed in higher layers. The formation of periodic structures has little effect on the phase transition temperature in the multilayer films. These structures can influence the functioning of spintronic elements implemented based on film structures on the substrate.

We will compare the results with some experimental facts. For thin films CrN on the Si substrate, a structural phase transition is observed with an increase in film thickness [24]. This phase transition is associated with a decrease in the effect of deformations on the film/substrate interface with an increase in film thickness. Experimental studies of a thin VO_2_ film on a sapphire substrate showed that its deformation depends on thickness [25]. The phase transition temperature also increases with increasing film thickness. An experimental study of highly deformed thin films BiFeO_3_ showed a significant effect of the substrate interface on the structure and structural phase transitions in them [26]. Increasing the film thickness reduces the dependence on the substrate. The state of the film becomes more stable.

Experimental studies of thin films Sm_0_._35_Pr_0_._15_Sr_0_._5_MnO_3_ on various single-crystal substrates confirm the conclusions about the change in Curie temperature [27]. The film undergoes compression deformation when deposited on the substrate LaAlO_3_. The temperature of the ferromagnetic phase transition rises to 165 K. The substrates SrTiO_3_ and La_0_._18_Sr_0_._82_ stretch the film. The Curie temperature drops to 120 K and 130 K, respectively. Thin films Sm_0_._53_Sr_0_._47_MnO_3_ on a single-crystal substrate LaAlO_3_ undergo compression deformations [28]. This compression leads to an increase in the phase transition temperature. An experimental study of thin films FeRh on substrates MgO, SrTiO_3_ and KTaO_3_ demonstrated the dependence of Curie temperature on deformations and film thickness [29]. Compression strains shift the Curie point above room temperature. Reducing the film thickness to less than 15 nm significantly reduces the phase transition temperature.

## Figures and Tables

**Figure 1 materials-15-02390-f001:**
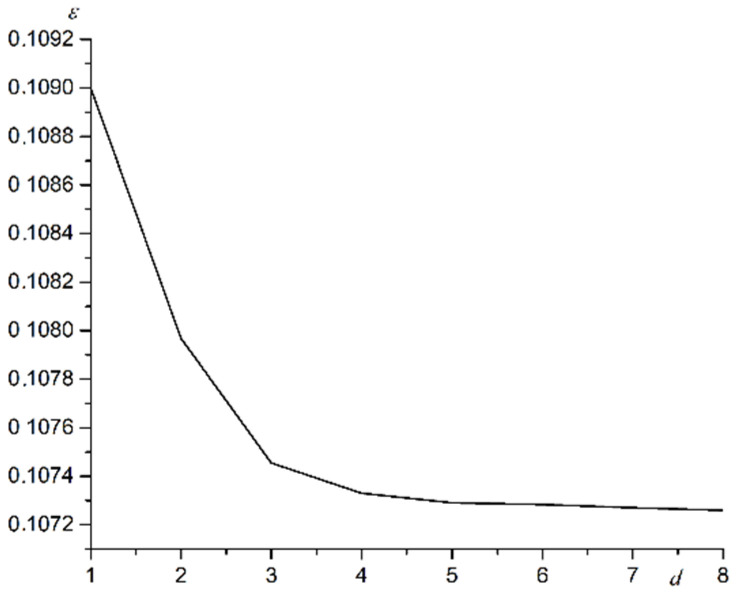
The relationship of the relative increase in the layer’s linear length *ε* depending on its number *d* at *b* = 1.10.

**Figure 2 materials-15-02390-f002:**
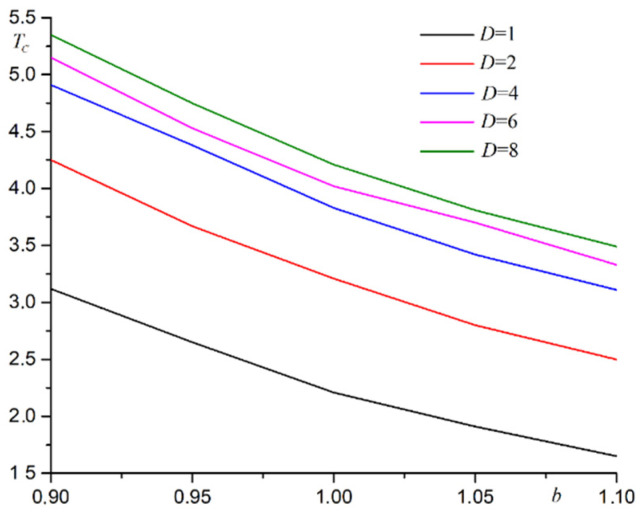
A plot of Curie temperature *T_C_* versus substrate period *b* for films with different thicknesses *D*.

**Figure 3 materials-15-02390-f003:**
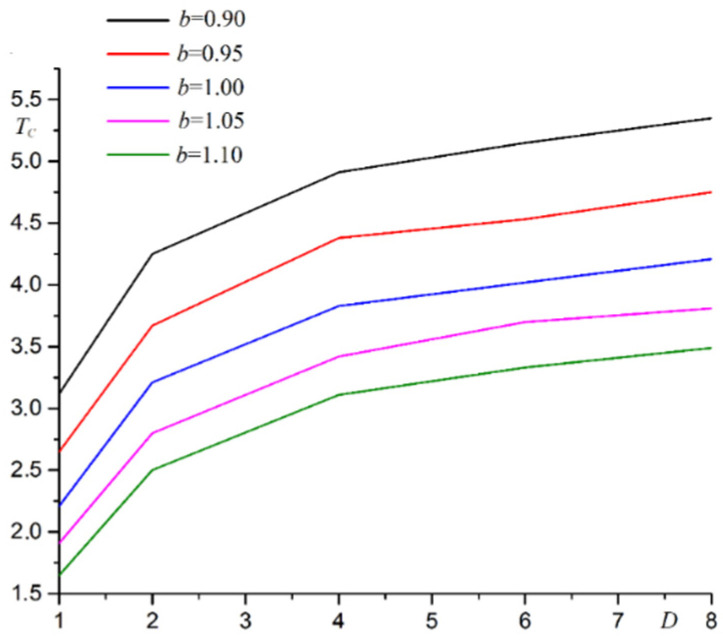
The dependence of Curie temperature *T_C_* on film thickness *D* at various substrate deformations *b*.

**Figure 4 materials-15-02390-f004:**
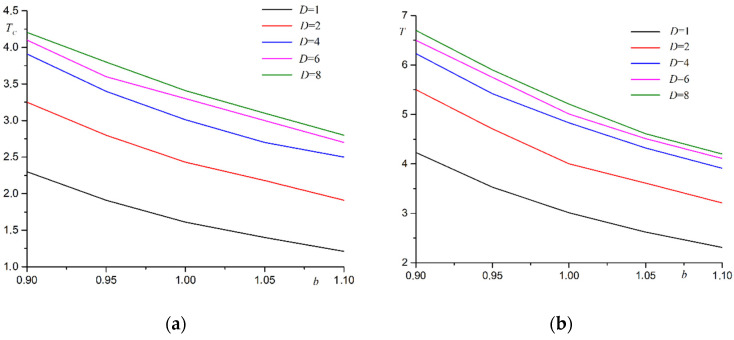
Dependence of Curie temperature *T_C_* on substrate period *b* for films with different thicknesses *D* at different coating coefficients: (**a**) *θ* = 0.9, (**b**) *θ* = 1.1.

**Figure 5 materials-15-02390-f005:**
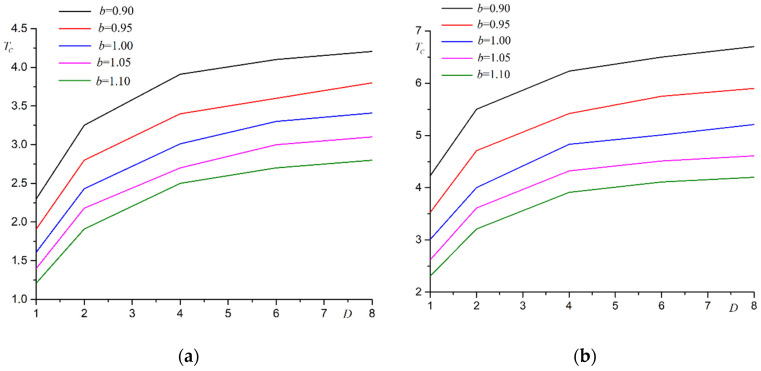
Dependence of phase transition temperature *T_C_* on film thickness *D*, at different substrate periods *b* at different coating coefficients: (**a**) *θ* = 0.9, (**b**) *θ* = 1.1.

**Figure 6 materials-15-02390-f006:**
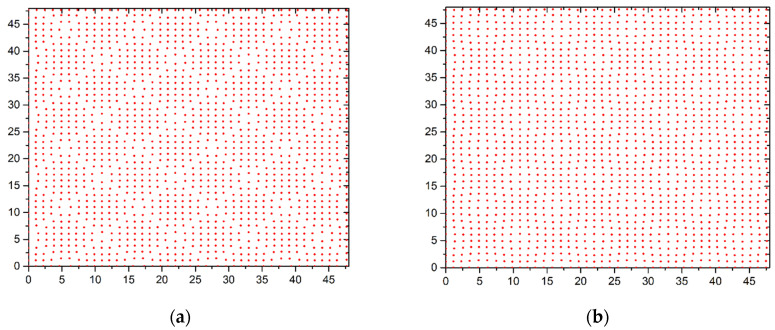
Placement of atoms in the first two layers at *θ* = 0.9 and *b* = 1.10: (**a**) the first layer, (**b**) the second layer.

**Figure 7 materials-15-02390-f007:**
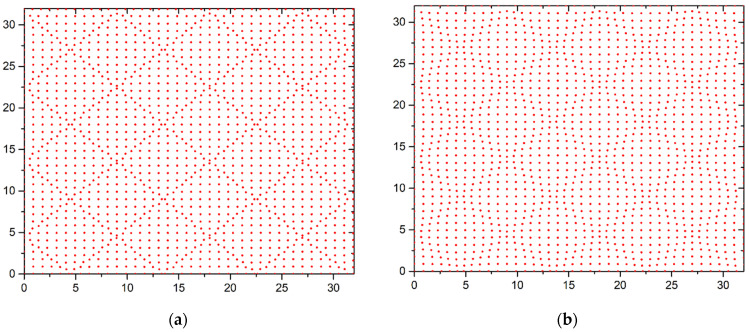
Placement of atoms in the first two layers at *θ* = 1.1 and *b* = 0.90: (**a**) the first layer, (**b**) the second layer.

## Data Availability

Not applicable.

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
