# Peer review of "Investigation of Phase Transitions in Ferromagnetic Nanofilms on a Non-Magnetic Substrate by Computer Simulation"

_materials, 2022, doi:10.3390/ma15072390_

Round 1

Reviewer 1 Report

Manuscript number: materials-1614898

Title: Investigation of phase transitions in ferromagnetic nanofilms on a non-magnetic substrate by computer simulation

In this paper, the authors examined the magnetic properties of ferromagnetic nanofilms on a non-magnetic substrate by computer simulation. The studied idea is interesting. Accordingly, I think that the scientific message is conveyed with clarity, which makes this publication of great value. The paper is acceptable for publication in Materials-MDPI, but it needs some rectifications and additions:

  1. The paper contains some grammatical errors and typo mistakes that should be corrected.
  2. The abstract should be improved by describing the interesting results and conclusions.
  3. Add more 2 keywords related to the work.
  4. A general introductory paragraph should be added at the beginning of the Introduction part.
  5. Why is a non-magnetic substrate selected? This should be more clarified and discussed in the Introduction part.
  6. The motivation and what will be done in the current study should be added at the end of the Introduction part.
  7. Also, if the substrate has a crystal lattice rather than a cubic one, what will be happened, and how this study could be helpful to other researchers? What parameters should be involved or altered? I think this will be important for readers and researchers.
  8. Check the symbols and equations.
  9. In the first paragraph of the results, what are the units of the parameters? Are they with unit or a.u.?
  10. The authors stated: “As seen in Figure 1, substrate deformations affect all layers. The substrate influence decreases as the number of the atomic layer increases.”. The authors should provide some explanations for that.
  11. What is the unit for the Curie temperature TC?
  12. Physical explanations should be provided on the variations of Curie temperature.

Author Response

Dear reviewer!

We thank you for your constructive comments. We provide answers to your questions and comments.

  1. The paper contains some grammatical errors and typo mistakes that should be corrected.

Answer: All noticed errors and typos corrected.

  1. The abstract should be improved by describing the interesting results and conclusions.

Answer: The abstract is supplemented by a description of the results.

  1. Add more 2 keywords related to the work.

Answer: I added keywords: computer simulation, Frenkel-Kontorova potential.

  1. A general introductory paragraph should be added at the beginning of the Introduction part.

Answer: Introductory paragraph added.

“Ferromagnetic film composite systems on a non-magnetic substrate are widely used in spintronics devices. A substrate of non-magnetic material has no effect on the giant magnetic resistance. The ability to control magnetization in such system is important. Antiferromagnetic or ferromagnetic substrates change the state in the magnetic film, creating an exchange bias. This effect is nonlinear. Non-magnetic substrates can change the mechanical properties of the film. Magnetostrictive effects change the magnetization of the film during mechanical deformations. The substrate deformations can cause a ferromagnetic phase transition at temperatures below the Curie point for the free film.”

  1. Why is a non-magnetic substrate selected? This should be more clarified and discussed in the Introduction part.

Answer: Explanations are given in the introductory paragraph.

  1. The motivation and what will be done in the current study should be added at the end of the Introduction part.

Answer: I added a paragraph at the end of the introduction.

Computer simulation of a thin ferromagnetic film on a substrate with a periodic potential is performed in this article. The simulation was carried out in two stages. In the first step, the size change for the film layers is examined by compressing and stretching the substrate. In the second step, the change in the magnetic properties for the film is studied under the influence of these deformations.

  1. Also, if the substrate has a crystal lattice rather than a cubic one, what will be happened, and how this study could be helpful to other researchers? What parameters should be involved or altered? I think this will be important for readers and researchers.

Answer: I added a paragraph.

“Crystal lattices with different geometry can be obtained by varying lattice periods along the OX and OY axes in potential (1). Examination of substrates with any crystal lattices can be carried out by adding periodic terms to the potential (1). The algorithm proposed in this paper can be used to model a system with any substrate potential. If the symmetry of the crystal lattice for the film and the substrate is different, then competition of the ordering types is observed in the system.”

  1. Check the symbols and equations.

Answer: Symbols and equations checked.

  1. In the first paragraph of the results, what are the units of the parameters? Are they with unit or a.u.?

Answer: The number of spin layers is usually indicated in computer simulations for such systems. Modeling is always done for relatively small systems. Computer simulations reveal general patterns. The correspondence of atomic layers in the computer model and real sizes in the experiment is an urgent problem that has not been completely solved in this field of physics.

  1. The authors stated: “As seen in Figure 1, substrate deformations affect all layers. The substrate influence decreases as the number of the atomic layer increases.”. The authors should provide some explanations for that.

Answer: I added a paragraph:

The first film layer interacts most intensively with the substrate. The deformations for the first layer almost exactly repeat the deformations for the substrate. Deformations are transmitted to the following layers by elastic forces of interaction between the film atoms. For layers with numbers greater than the first, there is a competition of elastic forces. The first force tends to deform the layer under the influence of the substrate. The second force tends to return the atoms to their original equilibrium state. The first force is created by layers located closer to the substrate. The second force is created by higher layers. The first force decreases as the distance to the substrate increases. Substrate deformations affect each next layer less.

  1. What is the unit for the Curie temperature TC?

Answer: The system temperature is measured in relative units.

“The temperature of system T is measured in relative units J0/kB (kB - Boltzmann constant) in computer modeling.”

  1. Physical explanations should be provided on the variations of Curie temperature.

Answer: I added a paragraph:

“The change in Curie temperature is caused by a change in the exchange integral for the interaction between spins. The exchange integral changes due to a decrease or increase in the distance between atoms possessing spins. The change in Curie temperature can be considered as a manifestation of the magnetostrictive effect during film deformation.”

Reviewer 2 Report

Paper ID # materials-1614898

The article looks interesting to me. However, I have a few comments to improve the quality of the manuscript. My comments are as follows:

Comments:

    1. The major results of this work need to be added to the conclusion.
    2. Also, the significance of this article was vaguely stated in the conclusion and introduction.
    3. In this work, the author demonstrated how the film thickness affects the magnetic properties of the materials. It would be interesting to see the experimental results to justify their claims.

Author Response

Dear reviewer!

We thank you for your constructive comments. We provide answers to your questions and comments.

  1. The major results of this work need to be added to the conclusion.

Answer: The paragraph is added at the end of the conclusion.

  1. Also, the significance of this article was vaguely stated in the conclusion and introduction.

Answer: Introduction and conclusion expanded.

  1. In this work, the author demonstrated how the film thickness affects the magnetic properties of the materials. It would be interesting to see the experimental results to justify their claims.

Answer: Descriptions and references to experimental results are added in conclusion.

Reviewer 3 Report

In this paper, the author investigated the phase transitions in ferromagnetic nanofilms on a non-magnetic substrate by computer simulation. By using Ising model and Metropolis algorithm, the dependence of Curie temperature on film thickness and substrate potential parameters was calculated. However, there are several issues the author should address before it is suitable for publication.

  1. In this work, the author investigated the impact of nonmagnetic substrate on magnetic properties of nanofilms. While the substrate-induced deformation plays a role, the influence of nonmagnetic/ferromagnetic interface can’t be neglected either, especially for nanofilms. The magnetic properties can be significantly modified by the chemical bonding at interface, which was extensively studied in last a few decades. Obviously, such an atomistic interface effect was ignored by the author. The author should clarify that issue in this manuscript.
  2. The author tried to solve a physical problem by performing computer simulations, but the setting of those simulation parameters seem arbitrary or nonrigorous. What is the relation or connection between the parameters and real materials? Perhaps, the author should specify a real nonmagnetic/ferromagnetic material system to perform the simulations.
  3. Equation (4) is somehow questionable. For some materials, the exchange coefficient J for the first nearest atom may have an opposite sign from the second one, i.e., the sign of J may be changed repeatedly with distance.
  4. There are many similar studies on this topic, so that the authors should search more and cite them in the manuscript. By the way, the texts should be checked carefully, because there are many grammar mistakes.

Author Response

Dear reviewer!

We thank you for your constructive comments. We provide answers to your questions and comments.

  1. In this work, the author investigated the impact of nonmagnetic substrate on magnetic properties of nanofilms. While the substrate-induced deformation plays a role, the influence of nonmagnetic/ferromagnetic interface can’t be neglected either, especially for nanofilms. The magnetic properties can be significantly modified by the chemical bonding at interface, which was extensively studied in last a few decades. Obviously, such an atomistic interface effect was ignored by the author. The author should clarify that issue in this manuscript.

Answer: The paragraph is added in the introduction:

“Different types of interactions may occur on the film and substrate interface. Chemical bonds at the interface between the film and the substrate play a large role in the formation of the film by various techniques. These bonds can determine the type of crystal lattice for the film. One type of crystal lattice is studied in this work, therefore, the main type of interaction by Van der Waals forces is considered. Different types of interactions should be considered when studying structural phase transitions induced by the substrate in nanofilms.”

  1. The author tried to solve a physical problem by performing computer simulations, but the setting of those simulation parameters seem arbitrary or nonrigorous. What is the relation or connection between the parameters and real materials? Perhaps, the author should specify a real nonmagnetic/ferromagnetic material system to perform the simulations.

Answer: The parameter a defines the basic distance scale. Parameter b is defined in units a, so a = 1 is used. The substrate deformations do not exceed 5% in a real experiment. Substrate deformations from 0% to 10% are studied in the article. The parameters of the exchange integral formula were studied in a wide range. The article presents the values of the most vividly demonstrating effects of the substrate influence.

  1. Equation (4) is somehow questionable. For some materials, the exchange coefficient J for the first nearest atom may have an opposite sign from the second one, i.e., the sign of J may be changed repeatedly with distance.

Answer: Formula (4) for the exchange integral describes a basic case of a rapid decrease in exchange interaction with distance. Considering the interaction with neighbors following the nearest ones leads to long-range effects. Changing the sign of the exchange integral requires considering competition between ferromagnetic and antiferromagnetic ordering. It is impossible to consider all these effects in one article. Long-range effects are usually considered as corrections to the model with an exponential decrease in the exchange integral.

  1. There are many similar studies on this topic, so that the authors should search more and cite them in the manuscript. By the way, the texts should be checked carefully, because there are many grammar mistakes.

Answer: The detected errors have been corrected. Conclusion expanded. Cites to work added.

Round 2

Reviewer 1 Report

The revised manuscript has been improved. The authors replied to the comments appropriately. I think that it can be now accepted for publication in Materials.

Author Response

Dear reviewer!
Thanks for the review. Spelling check performed. The errors found have been corrected.

Reviewer 2 Report

The authors made significant modifications to the revised manuscript. However, I have found multiple grammatical mistakes throughout the manuscript, which need to be taken care of.  It is advised to go through a thorough revision to resolve this issue.

Author Response

(The authors gave the same response as above.)

Reviewer 3 Report

The author has addressed most of my questions and the manuscript has been improved.

Author Response

(The authors gave the same response as above.)
